**Data Availability Statement:** All relevant data are within the paper and its Supporting information files.

**Funding:** This work was supported by Pfizer Inc. MG, LM and SB are employed by stève consultants

# A database study of clinical and economic burden of invasive meningococcal disease in France

Liping Huang[1], Stéphane Fievez[2], Mélanie Goguillot[3], Lucile Marié[3]*, Stève Bénard[3], Anne Elkaïm[2], Myint Tin Tin Htar[2]

**1** Pfizer, Collegeville, Pennsylvania, United States of America, **2** Pfizer France, Paris, France, **3** Stève Consultants, Oullins, France

* lucile.marieabc@gmail.com

## Abstract

### Objective

Invasive meningococcal disease (IMD) is life-threatening and associated with substantial morbidity and mortality. The study aimed to examine the clinical characteristics and hospital-based healthcare resource use and related costs following IMD diagnosis in France.

### Methods

Patients admitted to hospitals due to IMD between 2014 and 2016 were selected from the French hospital discharge database (PMSI). Demographics, clinical outcomes and health utilization (HRU) during index hospitalization were described. HRU and costs during the follow-up period were also examined. A generalized linear model was applied to examine 1-year costs after index hospitalization adjusting for age, type of IMD and presence of sequelae at index hospitalization.

### Results

A total of 1,344 patients were identified. About 30% cases were in children < 5 years old and 25% aged 10–24 years. Majority of patients presented as meningococcal meningitis (59%), 25% as meningococcaemia, and 9% both. The case fatality rate during the index hospitalization was 6%. About 15% of patients had at least one sequela at index hospital discharge. The median length of stay and the median cost of index hospitalization were 9 days and 8,045€, respectively. Patients with at least one sequela, with clinical manifestation as both meningitis and meningococcaemia, or aged 25 years and older were statistically significantly associated with higher costs than others.

### Conclusion

IMD is unpredictable and can occur in all ages. The study highlights the severity and high health and economic burdens associated with the disease. The data underlines the importance of prevention against IMD through vaccination.

which received funding from Pfizer Inc for the execution of the study and manuscript preparation. LH, SF, AE and MTTH are employees of Pfizer Inc.

**Competing interests:** The authors have read the journal's policy and the authors of this manuscript have the following competing interests: LH, SF, AE and MTTH are paid employees of Pfizer Inc. and hold company shares. LM, MG and SB are paid employees of Stève consultants, Oullins, France. This does not alter our adherence to PLOS ONE policies on sharing data and materials. There are no patents, products in development or marketed products to declare.

## Introduction

Invasive meningococcal disease (IMD) is an infectious disease caused by bacterium *Neisseria meningitidis*. In general, twelve different serogroups of N. meningitidis exist, but more than 96% of IMD cases are caused by serogroups A, B, C, Y, and W [1]. Despite availability of antibiotic treatment, IMD remains a serious public health concern because clinical symptoms are often nonspecific and progress rapidly. The case fatality rate ranged from 10% to 40% [2], 10% to 33% of survivors would develop complications being the highest rate observed in elderly [3, 4] and up to 20% of survivors experienced permanent physical, neurological, cognitive, behavioral and psychological disorders.

In France, the incidence of IMD is around 1/100,000. Between 2011–2018 period, a leading cause of IMD was serogroup B, followed by serogroup C. During 2014–2017, the incidence caused by serogroups W and Y had been increased with a slight decrease in 2018. But the number of W cases seemed to be on the rise again in the first half of 2019 with W representing 17% of total IMD cases of known serogroup in 2018/19 [5].

In order to prevent IMD, a routine meningococcal serogroup C conjugate (MenC) vaccine was introduced in all children at the age of 12 months with a catch-up vaccination up to the age of 24 years in 2010. However, due to low coverage among teenagers and young adults (68.2% for 2 years old, 22.5% for 15–19 years old and 9.4% for 20–24 years old in 2015), the incidence of serogroup C did not decrease but rather increased between 2010 and 2016, especially among the <1 year old. Thus since 2017, an additional dose of MenC vaccine at the age of 5 months to directly protect infants was recommended. The incidence of serogroup C has been decreased since 2017 [5].

Raising awareness and communicating evidence-based information on the importance of meningococcal vaccination is essential to increase the vaccination coverage to control IMD in France. Although IMD is a part of the infectious disease surveillance, limited data is available to describe clinical and economic burden of IMD. Therefore, this study aimed to examine the clinical characteristics as well as hospital-based healthcare resource utilizations and related costs following IMD diagnosis in France.

## Materials and methods

### Data source

The study was based on the French Medical Information System Program (PMSI) database. PMSI is a national database that covers all hospitals in public and private settings and includes patients records from the wards of medicine, surgery and obstetrics (MCO), and rehabilitation centers (SSR). The database includes information related to demographics, diagnoses, medical procedures, and healthcare utilization. All diagnoses were coded according to International Classification of Diseases, 10th revision (ICD-10). For each hospitalization, there are 3 different types of diagnoses: a primary diagnosis (PD) corresponding to the health condition leading to admission, a related diagnosis (RD) or a significant associated diagnosis (SAD) corresponding to a condition related to the PD or a new or existing comorbidity identified or treated.

Patients are classified through a standard discharge summary report into Diagnostic-Related Groups (DRG). Hospitalization costs were calculated based on the representative national cost study, the ENCC (*Echelle nationale des coûts à méthodologie commune*), which provides an average cost per DRG from a hospital perspective.

Data in PMSI are collected and hosted by the technical Agency for Information on Hospital Care (ATIH) and data were accessed via the platform of the Secure Data Access Center (CASD) after a declaration to the National Institute of Health Data (INDS) through the

reference methodology 006 (MR-006). Since all patient-level data in the PMSI database are anonymized, institutional review board/ethical approval and informed consent at an individual patient level was not required.

## Study design and study population

A retrospective study was conducted based on an IMD cohort identified from PMSI database. Patients who were hospitalized between January 1st, 2014 and December 31st, 2016 (study period), and had at least one IMD ICD-10 codes (S1 Table) as PD or had at least one ICD-10 codes related to IMD's manifestations as PD along with IMD ICD-10 codes as RD or SAD were included into the cohort.

The IMD cohort was further classified into the following mutually exclusive groups based on clinical manifestations defined by PD, RD or SAD: meningitis only (A39.0), septicaemia only (A39.2, A39.4), both septicaemia and meningitis, or other or unspecified meningococcal infection (A39.1, A39.5, A39.8, A39.9).

Index hospitalization was defined as the first hospitalization due to IMD and discharged during the study period. The follow-up period was defined as the duration between the index hospital discharge date to the end of the study period or death, whichever occurred first. The longest duration of follow-up period was 3 years.

Outcomes measured during the index hospitalization comprised the length of index hospitalization, admission to intensive care unit (ICU), procedures (mechanical ventilation, filing or use of catecholamine, dialysis for acute kidney injury), discharge status, including death, the presence of sequelae at discharge, and the direct medical cost. Outcomes measured during the follow-up period included all-cause rehospitalization, rehabilitation, presence of sequelae during the entire study period, and costs related to subsequent hospitalizations.

The pre-specified sequelae, including neurological sequelae (motor deficit, seizure, visual disturbance and hydrocephalus), hearing impairment, amputation or skin necrosis or skin grafting, were defined by the 2001 Global Burden of Disease Control Priorities Project (GBD DCPP) [6]. All sequelae were identified using algorithms based on ICD-10 codes as PD, RD or SAD and the procedure coding system in France (S2 Table).

## Statistical analysis

Continuous variables were summarized with standard descriptive statistics including means, standard deviations (SD), median, and interquartile (IQR). Categorical variables were summarized with frequencies and percentages. For cost associated with index hospitalization, actual cost as well as costs associated with each type of clinical manifestation were reported. For healthcare utilization during the follow-up period, the analyses were stratified based on number of patients within each segment of the follow-up period (0 to $\leq$ 6 months, 7–$\leq$ 12 month, 13–$\leq$ 24 months and 25–$\leq$ 36 months) and annualized costs were reported. These annualized costs have been calculated as average costs adjusted to the follow-up period and meant to compare costs estimated on different follow-up periods, by adjusting them by the follow-up period of patients.

Multivariate analysis was conducted in patients with at least one year of follow-up to examine factors associated with costs during the first year of IMD. Due to the statistical distribution of costs (costs are non-negative and tend to be skewed to the right), this analysis was derived from a generalized linear model with gamma distribution and a log link function. No model fitting tests was performed.

All costs were indexed in €2018 according to the index of the prices of health services consumption [7]. All statistical analyses were conducted using the Statistical Analysis System (SAS) statistical software package, version 9.3.

## Results

### Demographic and clinical characteristics of IMD cases

A total of 1,344 patients were identified and hospitalized for IMD in France between January 1st, 2014 to December 31th, 2016. An increase in the annual number of cases was observed: 410 in 2014, 448 in 2015 (+9%, compared with 2014) and 486 in 2016 (+8%, compared with 2015) (Fig 1).

Mean age at index hospitalization was 26.5 years (median = 19) and 46% were female (Table 1). About 30% of IMD cases occurred in children aged 4 years and younger and 25% of cases occurred in the age group of 10 to 24 years, whereas about 14% of the IMD cases affected persons aged 60 years and older (Table 1).

Concerning clinical manifestations, most of patients (58.8%) were hospitalized with meningitis only, followed by septicaemia only (25.1%) and both (9.4%). The lowest mean and median ages were observed in patients with clinical manifestation as both septicaemia and meningitis (mean = 18.5, median = 13), followed by patients with meningitis only (mean = 24.8, median = 19) (Table 1), while oldest patients were observed in the unspecified or other type of IMD group (mean = 37.9, median = 31). With respect of the age group, the highest proportion of patients with meningitis only was observed in the age group 15–19 years old (67.4%), followed by infants (62.5%) and adults aged 25–29 years old (62.0%). For patients with septicemia only, the highest proportion was observed in adults aged 65 years and older (34.0%), followed by young adults aged 20–24 years old (27.2%). The highest proportion with both types of manifestation was observed in the age group 10–14 years old (18.3%), followed by the age group 5–9 years old (17.9%).

Of total IMD patients, 44.9% were admitted to ICU, with the highest proportion in patients with both meningococcal septicaemia and meningitis (66.9%) and the lowest in patients with unspecified or other type of IMD (28.1%) (Table 2). During the index hospitalization, 31.5% of patients had at least one of procedures of interest. The common procedures performed were mechanical ventilation (22.5%) and filling or using of catecholamine (24.4%). Few patients also had kidney dialysis (3.3%).

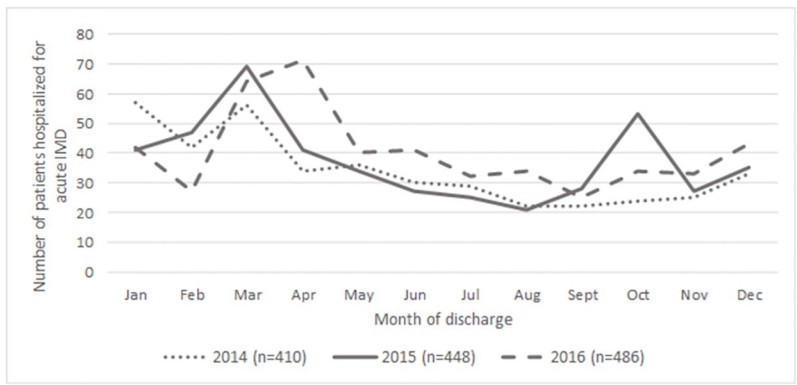

**Fig 1. Number of IMD cases identified during the study period (2014–2016).**

**Table 1. Characteristics of IMD patients by clinical manifestations during 2014–2016.**

|  | Septicaemia | Meningitis | Septicaemia and meningitis | Unspecified or other type of IMD | Total |
|---|---|---|---|---|---|
| **Total patients (%)** | **338 (25.1%)** | **790 (58.8%)** | **127 (9.4%)** | **89 (6.6%)** | **1,344** |
| Female (%) | 155 (45.9%) | 361 (45.7%) | 61 (48.0%) | 39 (43.8%) | 616 |
| Age |  |  |  |  |  |
| Mean (SD) | 30.3 (29.2) | 24.8 (24.4) | 18.5 (20.7) | 37.9 (29.6) | 26.5 (26.1) |
| Median (IQR) | 20 (3–52) | 19 (3–41) | 13 (2–22) | 31 (11–65) | 19 (3–46) |
| Age Group: N (%) |  |  |  |  |  |
| < 1 year | 50 (24.0%) | 130 (62.5%) | 18 (8.7%) | 10 (4.8%) | 208 |
| 1–4 years | 49 (25.5%) | 113 (58.9%) | 23 (12.0%) | 7 (3.6%) | 192 |
| 5–9 years | 18 (23.1%) | 42 (53.8%) | 14 (17.9%) | 4 (5.1%) | 78 |
| 10–14 years | 12 (16.9%) | 41 (57.7%) | 13 (18.3%) | 5 (7.0%) | 71 |
| 15–19 years | 23 (17.0%) | 91 (67.4%) | 16 (11.9%) | 5 (3.7%) | 135 |
| 20–24 years | 37 (27.2%) | 77 (56.6%) | 14 (10.3%) | 8 (5.9%) | 136 |
| 25–59 years | 80 (24.9%) | 199 (62.0%) | 17 (5.3%) | 25 (7.8%) | 321 |
| ≥ 60 years | 69 (34.0%) | 97 (47.8%) | 12 (5.9%) | 25 (12.3%) | 203 |

IMD: Invasive meningococcal disease; IQR: Interquartile Range; SD: Standard Deviation.

The overall case fatality rate (CFR) during the index hospitalization was 6.0% (n = 80) with highest CFR observed in patients with clinical manifestation as septicaemia only (7.7%) and the lowest CFR in patients with both (3.9%). Majority of patients were discharged to home (79.1%) and 5.0% of patients were transferred to rehabilitation at discharge from the index hospitalization.

Of the total study population, 13.6% (n = 183) patients had at least one sequela at index hospital discharge. The highest proportion of patients with at least one sequela was patients with both septicaemia and meningitis (16.5%), followed by patients with meningitis only (13.9%) and septicaemia(12.4%). Neurological sequelae occurred in 8.1% (n = 109) of IMD cases and was more frequent in patients with meningitis (10.0%) or unspecified IMD (10.1%) than in patients with both meningitis and septicaemia (7.9%) or septicaemia only (3.3%). Amputation or skin necrosis or skin grafting occurred in 2.7% (n = 36) of patients. Other sequelae occurred in less than 5% of patients.

With regard to the outcomes during the entire study period, overall about one-fifth of patients developed at least one sequela (19.4%). Compared with numbers observed at index hospital discharge (13.6%), about 30% of sequelae were developed after discharge from the index hospitalization. Notably, about 70% of cognitive impairment was diagnosed during the follow-up period (16 of 23). The distribution of sequelae in relation to clinical manifestations during the entire study period is similar to the distribution during the index hospitalization. For example, the highest proportion of patients with at least one sequela during the entire study period was observed in patients with both septicaemia and meningitis (21.3%), followed by patients with meningitis only (19.9%) and septicaemia only (18.3%).

## Healthcare resource use and attributable costs

**Index hospitalization.** The overall mean and median length of index hospitalization stay were 12.0 days (SD = 10.9) and 9 days (Q1 = 7 and Q3 = 13 days), respectively, and were similar across different types of clinical manifestations (Table 2).

The mean direct medical cost of the index hospitalization was €11,445 (SD = €9,988) and the median cost was €8,045 (IQR: €5,233-€12,714). Patients with both septicaemia and

**Table 2. Clinical characteristics during the index hospitalization and the entire study period.**

| | Septicaemia Only | Meningitis only | Septicaemia and Meningitis | Unspecified or other type of IMD | Total |
|---|---|---|---|---|---|
| **Number of patients (%)** | 338 (25.1%) | 790 (58.8%) | 127 (9.4%) | 89 (6.6%) | 1,344 |
| **During the Index Hospitalization** | | | | | |
| **Length of Index Hospital Stay (days)** | | | | | |
| *Mean (SD)* | 12.3 (12.9) | 11.5 (9.2) | 13.3 (13.4) | 12.9 (13.2) | 12.0 (10.9) |
| *Median (Q1-Q3)* | 9 (6–14) | 9 (7–13) | 9 (8–14) | 9 (6–15) | 9 (7–13) |
| **ICU admission, N (%)** | 140 (41.4%) | 354 (44.8%) | 85 (66.9%) | 25 (28.1%) | 604 (44.9%) |
| **Procedures of interest, N (%)** | 120 (35.5%) | 218 (27.6%) | 68 (53.5%) | 17 (19.1%) | 423 (31.5%) |
| *Mechanical ventilation* | 77 (22.8%) | 174 (22.0%) | 41 (32.3%) | 11 (12.4%) | 303 (22.5%) |
| *Filing/use of catecholamine* | 114 (33.7%) | 141 (17.8%) | 60 (47.2%) | 13 (14.6%) | 328 (24.4%) |
| *Dialysis for acute kidney injury* | 20 (5.9%) | 17 (2.2%) | 7 (5.5%) | 1 (1.1%) | 45 (3.3%) |
| **Discharge status, N (%)** | | | | | |
| *Death* | 26 (7.7%) | 44 (5.6%) | 5 (3.9%) | 5 (5.6%) | 80 (6.0%) |
| *Rehabilitation center* | 21 (6.2%) | 31 (3.9%) | 6 (4.7%) | 9 (10.1%) | 67 (5.0%) |
| *Home* | 253 (74.9%) | 641 (81.1%) | 102 (80.3%) | 67 (75.3%) | 1,063 (79.1%) |
| *Other* | 38 (11.2%) | 74 (9.4%) | 14 (11.0%) | 8 (9.0%) | 134 (10.0%) |
| **With at least one sequela at index hospital discharge, N (%)** | | | | | |
| *Any* | 42 (12.4%) | 110 (13.9%) | 21 (16.5%) | 10 (11.2%) | 183 (13.6%) |
| *Neurological Sequelae* | 11 (3.3%) | 79 (10.0%) | 10 (7.9%) | 9 (10.1%) | 109 (8.1%) |
| *Auditive impairment* | 0 (0.0%) | 26 (3.3%) | 1 (0.8%) | 0 (0.0%) | 27 (2.0%) |
| *Cognitive impairment* | 4 (1.2%) | 3 (0.4%) | 0 (0.0%) | 0 (0.0%) | 7 (0.5%) |
| *Chronic renal failure* | 8 (2.4%) | 4 (0.5%) | 2 (1.6%) | 2 (2.2%) | 16 (1.2%) |
| *Amputation/ Skin necrosis/Skin grafting* | 19 (5.6%) | 8 (1.0%) | 8 (6.3%) | 1 (1.1%) | 36 (2.7%) |
| *Arthritis* | 5 (1.5%) | 3 (0.4%) | 2 (1.6%) | 0 (0.0%) | 10 (0.7%) |
| **With at least one sequela during the entire study period, N (%)** | | | | | |
| *Any* | 62 (18.3%) | 157 (19.9%) | 27 (21.3%) | 15 (16.9%) | 261 (19.4%) |
| *Neurological Sequelae* | 19 (5.6%) | 113 (14.3%) | 11 (8.7%) | 11 (12.4%) | 154 (11.5%) |
| *Auditive impairment* | 1 (0.3%) | 34 (4.3%) | 2 (1.6%) | 0 (0.0%) | 37 (2.8%) |
| *Cognitive impairment* | 6 (1.8%) | 17 (2.2%) | 0 (0.0%) | 0 (0.0%) | 23 (1.7%) |
| *Chronic renal failure* | 11 (3.3%) | 6 (0.8%) | 4 (3.1%) | 4 (4.5%) | 25 (1.9%) |
| *Amputation/ Skin necrosis/Skin grafting* | 25 (7.4%) | 14 (1.8%) | 10 (7.9%) | 3 (3.4%) | 52 (3.9%) |
| *Arthritis* | 9 (2.7%) | 8 (1.0%) | 2 (1.6%) | 0 (0.0%) | 19 (1.4%) |

meningitis incurred the highest cost (mean = €13,782, median = €10,589), followed by patients with meningococcal meningitis only (mean = €12,103, median = €7,425). Patients with other or unspecified IMD had the lowest cost (mean = €8,679, median = €7,180) (Fig 2).

**Post-index hospitalization.** A total of 1,264 (94.0%) patients were survived from the index hospitalization and had at least 1 day of follow-up time (Table 3). The average duration of follow-up time of these patients was 17.1 months (median = 17 months). Of them, 1,019 (80.6%) had at least more than 7 months of follow-up time, 760 (60.1%) had at least 13 months, and 340 (26.9%) had at least 25 months of follow-up time.

Overall, of all patients survived from the index hospitalization, 41.8% had rehospitalization for any causes and 9.3% had at least one rehabilitation stay during the follow-up time. Rehospitalizations and rehabilitation were mainly observed during the first six months after discharge from the index hospitalization (33.5% and 8.5%, respectively); noteworthy, the proportions of

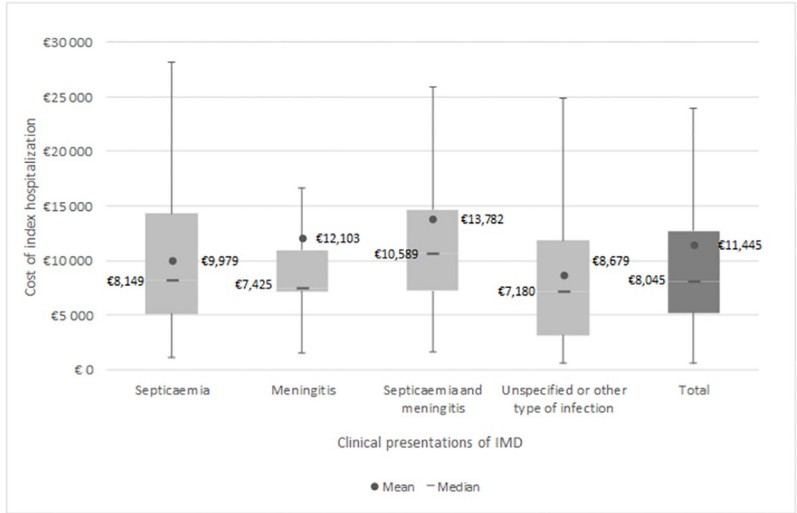

**Fig 2. Cost of the index hospitalization.**

patients with rehospitalization or rehabilitation after 6 months of the follow-up time were much lower than the proportions during the first 6 months and remained relatively stable thereafter. Patients with at least one sequela at index hospital discharge had higher proportions of rehospitalization or rehabilitation than patients without any sequelae (64.7% vs 36.2% and 27.3% vs 4.8%, respectively) (Table 3).

The average total actual direct medical cost over the follow-up period was €5,355 (median = €0) (Table 3), and it was higher in patients with at least one sequela at the index hospital discharge (mean = €15,151, median = €3,502) than patients without any sequelae (mean = €2,952, median = €0).

The means actual direct medical costs within the first 6 months, 7–12 months, 13–24 months, and 25–36 months after the index hospital discharge were €3,664, €986, €1,224, and €593, respectively. After adjusted for duration of observation time within each time period, the mean annualized costs were estimated at €9,549, €2,295, €2,407 and €881, respectively. Compared patients with and without any sequelae, in general the mean actual costs during different follow-up periods were higher in patients with at least one sequela than in patients without any sequelae, and the greatest difference was observed during the first 6 months after the index hospital discharge (actual cost mean: €10,731 vs. €1,931, median: €1,887 vs. €0) (Table 3).

Further examination in patients with at least 12 months of follow up (n = 792) showed that the average actual cost within 1-year of follow-up time was €16,832 (SD = €20,855, median = €10,589, IQR: €7,180-€17,182). Results from the generalized model indicated that the 1-year cost in patients with at least one sequela was 2.48 times higher than patients without sequelae (95% CI: 2.20–2.83) (Table 4). Compared with patients with clinical manifestation as meningitis only, the cost was 0.84 (95% CI: 0.74–0.95) and 0.75 (95% CI; 0.61–0.92) times lower in patients with septicaemia only or with other or unspecified clinical type, respectively, whereas it was 1.11 (95% CI: 0.92–1.33) times higher in patients with both septicaemia and meningitis (Table 4). Using the age group < 1 year as the reference group, the 1-year cost in patients aged between 25 and 59 years old or older than 59 years were 1.62 (95% CI: 1.37–1.90) and 1.55 (95%CI: 1.28–1.88) times higher than the reference group, respectively, whereas it was 0.83 (95% CI: 0.69–0.99) times lower in patients aged between 1 to 4 years old. If compared the

**Table 3. Healthcare utilization over the follow-up period.**

| | 0–≤6 months | 7–≤12 months | 13–≤24 months | 25–<36 months | Total |
|---|---|---|---|---|---|
| **Patients survived after the index hospitalization** | 1,264 (100%) | 1,019 (80.6%) | 760 (60.1%) | 340 (26.9%) | 1,264 (100%) |
| **N (% of total survived)** | | | | | |
| Follow-up (month) | | | | | |
| *Mean (SD) during a time period* | 5.4 (1.4) | 5.3 (1.5) | 9.2 (3.6) | 7.1 (3.1) | 17.1 (10.6) |
| *Median (Q1-Q3) during a time period* | 6 (6–6) | 6 (6–6) | 11 (7–12) | 8 (5–10) | 17 (8–26) |
| All-cause rehospitalization, *N (%)* | 424 (33.5%) | 136 (13.3%) | 123 (16.2%) | 41 (12.1%) | 528 (41.8%) |
| Rehabilitation, *N (%)* | 108 (8.5%) | 20 (2.0%) | 10 (1.3%) | 3 (0.9%) | 117 (9.3%) |
| Actual Costs during a time period, | | | | | |
| *Median (Q1-Q3)* | 0 (0–1,448) | 0 (0–0) | 0 (0–0) | 0 (0–0) | 0 (0–2,331) |
| *Mean (SD)* | 3,664 (11,054.3) | 986 (5,339.6) | 1,224 (6,673.9) | 593 (3,175.6) | 5,355 (17,857.1) |
| Annualized Costs of a time period, | | | | | |
| *Median (Q1-Q3)* | 0 (0–3,574) | 0 (0–0) | 0 (0–0) | 0 (0–0) | 0 (0–2,202) |
| *Mean (SD)* | 9,549 (29,530.4) | 2,295 (12,608.5) | 2,407 (17,114.0) | 881 (4,603.7) | 6,752 (24,656.2) |
| **Patients survived without any sequela at index hospital discharge** | 1,015 (80.3%) | 809 (79.4%) | 598 (78.7%) | 270 (79.4%) | 1,015 (80.3%) |
| **N (% of total survived)** | | | | | |
| Follow-up (month) | | | | | |
| *Mean (SD) during a time period* | 5.4 (1.5) | 5.3 (1.5) | 9.2 (3.6) | 7.1 (3.1) | 16.9 (10.7) |
| *Median (IQR) during a time period* | 6 (6–6) | 6 (6–6) | 11 (7–12) | 8 (5–10) | 16 (8–26) |
| All-cause rehospitalization, *N (%)* | 282 (27.8%) | 87 (10.8%) | 79 (13.2%) | 26 (9.6%) | 367 (36.2%) |
| Rehabilitation, *N (%)* | 45 (4.4%) | 8 (1.0%) | 4 (0.7%) | 1 (0.4%) | 49 (4.8%) |
| Actual Costs during a time period, | | | | | |
| *Median (IQR)* | 0 (0–826) | 0 (0–0) | 0 (0–0) | 0 (0–0) | 0 (0–1,246) |
| *Mean (SD)* | 1,931 (6,702.3) | 522 (3,004.9) | 822 (5,335.2) | 454 (2,900.9) | 2,952 (11,044.8) |
| Annualized Costs of a time period, | | | | | |
| *Median (IQR)* | 0 (0–1,700) | 0 (0–0) | 0 (0–0) | 0 (0–0) | 0 (0–1,051) |
| *Mean (SD)* | 5,512 (21,354.3) | 1,216 (7,316.4) | 1,469 (11,820.7) | 692 (4,194.5) | 4,212 (19,438.4) |
| **Patients survived with at least one sequela at index hospital discharge** | 249 (19.7%) | 210 (20.6%) | 162 (21.3%) | 70 (20.6%) | 249 (19.7%) |
| **N (% of total survived)** | | | | | |
| Follow-up (month) | | | | | |
| *Mean (SD) during a time period* | 5.6 (1.3) | 5.3 (1.4) | 9.2 (3.5) | 6.9 (3.3) | 18.0 (10.3) |
| *Median (Q1-Q3) during a time period* | 6 (6–6) | 6 (6–6) | 11 (7–12) | 8 (5–10) | 19 (9–26) |
| All-cause rehospitalization, *N (%)* | 142 (57.0%) | 49 (23.3%) | 44 (27.2%) | 15 (21.4%) | 161 (64.7%) |
| Rehabilitation, *N (%)* | 63 (25.3%) | 12 (5.7%) | 6 (3.7%) | 2 (2.9%) | 68 (27.3%) |
| Actual Costs within a time period, | | | | | |
| *Median (Q1-Q3)* | 1,887 (0–12,404) | 0 (0–617) | 0 (0–705) | 0 (0–0) | 3,502 (0–17,842) |
| *Mean (SD)* | 10,731 (19,396.0) | 2,774 (9,996.0) | 2,709 (10,080.3) | 1,130 (4,046.0) | 15,151 (31,704.6) |
| Annualized Costs of a time period, | | | | | |
| *Median (Q1-Q3)* | 3,887 (0–31,277) | 0 (0–1,234) | 0 (0–758) | 0 (0–0) | 2,468 (0–15,823) |
| *Mean (SD)* | 26,002 (47,309.0) | 6,452 (23,356.5) | 5,870 (29,108.5) | 1,609 (5,904.4) | 17,104 (37,644.5) |

reference group to other age groups (5–9 years, 10–14 years, 15–19 years, 20–25 years), statistically significant differences were not found (Table 4).

## Discussion

To our knowledge, this study is the first study in France to use the electronic database to assess the consequence of IMD. It analyzed IMD cases recorded in the PMSI from 2014 to 2016 and

**Table 4. Determinants of 1-year costs (€ 2018) among patients with at least 1-year follow-up (n = 792).**

| Determinants | Cost ratio | 95% confidence interval |
|---|---|---|
| **Presence of Sequela at Index Hospital Discharge** | | |
| No sequelae (reference) | 1.00 | - |
| At least one sequela | 2.48 | (2.20–2.83) |
| **IMD Clinical Manifestations** | | |
| Meningitis only (reference) | 1.00 | - |
| Septicaemia only | 0.84 | (0.74–0.95) |
| Septicaemia and meningitis | 1.11 | (0.92–1.33) |
| Other or unspecified meningococcal infection | 0.75 | (0.61–0.92) |
| **Age Group** | | |
| < 1 year (reference) | 1.00 | |
| 1–4 years | 0.83 | (0.69–0.99) |
| 5–9 years | 0.95 | (0.75–1.21) |
| 10–14 years | 0.88 | (0.68–1.12) |
| 15–19 years | 1.01 | (0.82–1.24) |
| 20–24 years | 1.03 | (0.84–1.27) |
| 25–59 years | 1.62 | (1.37–1.90) |
| ≥ 60 years | 1.55 | (1.28–1.88) |

assessed sequelae occurred at index hospital discharge and during a long-term observation period. In addition, it also examined health resource utilization and direct medical costs associated with IMD. Overall, the study identified 1,344 IMD cases and showed an increasing trend from 2014 to 2016. The numbers reported from this study were not far from the numbers reported in the national surveillance system for IMD (total 1401 cases from 2014 to 2016, with 417 in 2014, 462 in 2015 and 522 cases in 2016 [8–11]).

From our study, off all IMD cases, 30% of cases occurred in children <5 years old and 25% in 10 to 24 years old. Our study confirmed that infants, and young children, adolescents and young adults are at the highest risk of being infected with IMD. With regard to the clinical manifestations, our study showed that meningitis is the predominant clinical manifestation especially in adolescents 15–19 years old, in which about two-third of patients presented with meningitis only. Septicaemia was more frequent in older age groups and about one-third of patients aged 60 years and older presented with septicaemia. However the highest proportion of patients with both meningitis and septicaemia were observed in the age group 10–14 years old. The distribution of IMD clinical manifestations reported from our study is similar to what were reported previously. In a study in the United States before the introduction of meningococcal vaccination program in adolescents, the meningitis and septicemia represented 70% and 27% respectively of all IMD cases in children [12].

IMD is associated with significant clinical burdens. First, the study showed that about 6% of patients died during the index hospitalization, which is within the range of a recent review which reported a case fatality rate ranged from 4 to 20%, varying across serogroups and age groups [13]. Second, among all survivors, 19% of the patients reported at least one sequela during the entire study period and about 7% experienced delayed occurrence of sequela after the index hospitalization. In a literature published in 2010, it described that median risk of having at least one major sequela from meningococcal meningitis was 7.2% and 2.9% had cognitive difficulties [14]. A cohort study conducted in the Netherlands reported that, after a mean follow-up period of 15 years, about 35% of patients survived from meningococcal septic shock had developed at least one sequela [15]. The study period of our study was shorter than the

one conducted in the Netherlands and was only based on certain pre-defined sequelae, and thus some sequelae occurred long-termly might be missed in our study.

IMD is also associated with significant economic burdens. Wang et al. reported acute admission costs per patient based on international dollars (I) and results was ranged from I $1,629 in Colombia to I$50,796 in USA, with I$9,182 in Belgium and I$14,855 in Australia [16]. The mean cost of the index hospitalization from our study was €11,445 (i.e. I$14,306, converted to international dollar [17]) and is within the range of published estimates. The mean cost during the entire follow-up period was €5,355 and most of follow-up costs occurred during the first 6 month after discharged from the index hospitalization and was much higher among patients with at least one sequela. The average 1-year cost of €16,832 was reported from our study, which is close to the 1-year cost observed from Danish registries (i.e. I$14,817) [18] but is lower than 1-year costs observed in the US reported by Davis et al. [19] (I$50,796) and Karve et al. [20] (I$45,162). However, even if all costs can be converted into international dollars, studies from different countries cannot be directly compared due to differences in economic environments, healthcare policies, as well as study designs and definitions of cost. Results from the generalized model showed that 1-year costs in patients with at least one sequela was 2.48 times higher than the costs in patients without sequelae. Literature reported cost ratio for the presence of sequelae ranged from 1.91 to 4.28 [19–21]. Karve et al. [20] showed that the total healthcare costs in patients with sequelae was 2.98 times higher than the costs in patients without sequelae.

The main strength of the study was that, using the national database, the study captured majority of IMD cases during the study period in France. Over 60% of patients included in this study were retrospectively followed for at least one year and 27% for at least 2 years, allowing sequelae occurred during the latent follow-up periods to be gathered.

However, there are some limitations. First, the IMD cases and sequelae were identified based on ICD-10 codes. We could not completely rule out possibility of misclassification. Second, patients did not have the same follow-up period which limited equal opportunity to observe the occurrence of the sequelae or utilize healthcare services. In addition, it also indicated that the study follow-up time might not be long enough to capture all sequelae as it may take more than 3 years to confirm diagnoses of permanent neurological and psychological sequelae in infant survivors [16]. Finally, the study only considered the direct inpatient medical cost. Direct medical costs associated with other healthcare settings or indirect costs associated with caregivers or patients, (e.g. handicap compensation and costs associated with sick leave and absenteeism form work for taking care patients), costs of public health response for an outbreak (chemoprophylaxis and contact tracing) were not included. However, the cost of acute treatment and readmissions is the most important components of total healthcare costs [16].

## Conclusions

IMD is associated with severe complications, sequelae, and costs. While most of the cases occurred in children under 5 years of age, considerable numbers of cases occurred in adolescents and adults over 65 years of age. The study highlights the extensive costs related to IMD especially during the early stage. Sequelae occurred during latent period in children, such as auditive and cognitive impairments, could also have considerable impact in patients, caregivers and societies. Overall, the study underscored that IMD is associated with substantial clinical and economic burdens and it is important to raise awareness of the disease and to increase vaccination coverage to protect against IMD.

## Supporting information

**S1 Table. ICD-10 codes used to identify the study population.**
(DOCX)

**S2 Table. ICD-10 codes used to identify IMD-related sequelae.**
(DOCX)

## Acknowledgments

We would like to thank to CASD—Centre d'accès sécurisé aux données (Ref. 10.34724/CASD)
for making accessible to some confidential data within a secured environment.

## Author Contributions

**Conceptualization:** Liping Huang, Stéphane Fievez, Mélanie Goguillot, Stève Bénard, Anne
Elkaïm, Myint Tin Tin Htar.

**Formal analysis:** Mélanie Goguillot.

**Methodology:** Liping Huang, Stéphane Fievez, Mélanie Goguillot, Lucile Marié, Stève Bénard,
Anne Elkaïm, Myint Tin Tin Htar.

**Project administration:** Lucile Marié.

**Software:** Mélanie Goguillot.

**Supervision:** Liping Huang.

**Writing – original draft:** Lucile Marié.

**Writing – review & editing:** Liping Huang, Stéphane Fievez, Mélanie Goguillot, Stève Bénard,
Anne Elkaïm, Myint Tin Tin Htar.

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
