## [Decision Letter · Decision Letter 0]

1 Sep 2021

PONE-D-20-34991

A database study of clinical and economic burden of IMD in France

PLOS ONE

Dear Dr. Marié

Thank you for submitting your manuscript to PLOS ONE. After careful consideration, we feel that it has merit but does not fully meet PLOS ONE’s publication criteria as it currently stands. Therefore, we invite you to submit a revised version of the manuscript that addresses the points raised during the review process.

Please submit your revised manuscript by 01 October 2021. If you will need more time than this to complete your revisions, please reply to this message or contact the journal office at plosone@plos.org. Please include the following items when submitting your revised manuscript:

We look forward to receiving your revised manuscript.

Kind regards,

Silvia Ricci

Academic Editor

PLOS ONE

Journal Requirements:

3. Thank you for providing the following Funding Statement: 

“This project was funded by Pfizer France. LH, SF, AE and MTTH are employees of Pfizer. They had a role in study design, decision to publish and preparation of the manuscript.

SB, LM and MG were paied by Pfizer France as a consultancy company for study design, data collection in analysis, and development of the manuscript.”

We note that one or more of the authors is affiliated with the funding organization, indicating the funder may have had some role in the design, data collection, analysis or preparation of your manuscript for publication; in other words, the funder played an indirect role through the participation of the co-authors.

If the funding organization did not play a role in the study design, data collection and analysis, decision to publish, or preparation of the manuscript and only provided financial support in the form of authors' salaries and/or research materials, please review your statements relating to the author contributions, and ensure you have specifically and accurately indicated the role(s) that these authors had in your study in the Author Contributions section of the online submission form. Please make any necessary amendments directly within this section of the online submission form.  Please also update your Funding Statement to include the following statement: “The funder provided support in the form of salaries for authors [insert relevant initials], but did not have any additional role in the study design, data collection and analysis, decision to publish, or preparation of the manuscript. The specific roles of these authors are articulated in the ‘author contributions’ section.”

If the funding organization did have an additional role, please state and explain that role within your Funding Statement.

Please also provide an updated Competing Interests Statement declaring this commercial affiliation along with any other relevant declarations relating to employment, consultancy, patents, products in development, or marketed products, etc. 

Reviewers' comments:

Reviewer's Responses to Questions

**Comments to the Author**

1. Is the manuscript technically sound, and do the data support the conclusions?

Reviewer #1: Yes

2. Has the statistical analysis been performed appropriately and rigorously? 

Reviewer #1: Yes

3. Have the authors made all data underlying the findings in their manuscript fully available?

Reviewer #1: Yes

4. Is the manuscript presented in an intelligible fashion and written in standard English?

Reviewer #1: Yes

5. Review Comments to the Author

Reviewer #1: Thanks so much for inviting me to review this manuscript. It is well written and very interesting paper. Vaccination has greatly reduced the burden of infectious diseases. It is a double edged sword. Vaccine-preventable diseases have gone down, followed by a rise in vaccine hesitancy. It is great to read a paper investigating the burden of IMD, a life threatening disease. I only have a few minor comments.

Financial disclosure (page 1): In the Acknowledgement section, the authors stated the work was supported by a public grant. This grant was not mentioned here and it was only said that the project was funded by Pfizer.

Introduction

Line 31: The authors stated that case fatality rate (CRF) ranged from 10% to 40%. Only in elderly people, the CFR could reach 40%. On average, the CFR ranged from 4-20%.

Statistical Analysis

Line 102: can the author please add more information and explain why GLM with gamma distribution and a log link function was used? Did the authors conduct any model fitting tests?

Results

Lines 117/120/126: Error messages. Can the authors remove or revise the references?

Table 2: Amputation/skin necrosis. I would expect amputation and skin necrosis could all be diagnosed during the index hospitalisation. But, 16 patients with amputation and skin necrosis were identified after the index hospitalisation. I suspect a few skin grafting procedures might be performed after the index hospitalisation. If that was true, can the authors please add skin grafting?

Line 155: Would it better to say numbers (16 of 23) rather than percentages (1.2% of 1.7%)?

Table 4: The annualised costs were reported. Can the authors please explain in the statistical analysis section how the annualised costs were calculated?

Table 5: The total number of patients with at least one-year follow up is 792. However, Table 4 says 760 had at least one-year follow up time. Can the authors explain why 32 patients were not included in Table 4?

Discussion

Line 270: survivors?

Limitations: Serogroups could not be included in the analysis. Were outpatient visits included? For IMD cases with limb disabilities, speech problems and chronic kidney impairments, physiotherapy, speech therapy and dialysis costs could be high as well.

6. PLOS authors have the option to publish the peer review history of their article (what does this mean?). If published, this will include your full peer review and any attached files.

Reviewer #1: No

---

## [Author Response · Author response to Decision Letter 0]

10 Nov 2021

All comments from the Journal and from the reviewers have been answered in the response to reviewers. 

The financial disclosure has been added in the manuscript and the funding information has been updated. 

No change has been made on the references.

For data availability, there is some restrictions to access data. The data is available within a secured environment making accessible by CASD – Centre d’accès sécurisé aux données (Ref. 10.34724/CASD). No data export is authorized.

---

## [Decision Letter · Decision Letter 1]

18 Apr 2022

A database study of clinical and economic burden of Invasive Meningococcal Disease in France

PONE-D-20-34991R1

Dear Dr. Marié,

We’re pleased to inform you that your manuscript has been judged scientifically suitable for publication and will be formally accepted for publication once it meets all outstanding technical requirements.

Kind regards,

Carla Pegoraro

Division Editor

PLOS ONE

Additional Editor Comments (optional):

Reviewers' comments:

Reviewer's Responses to Questions

**Comments to the Author**

1. If the authors have adequately addressed your comments raised in a previous round of review and you feel that this manuscript is now acceptable for publication, you may indicate that here to bypass the “Comments to the Author” section, enter your conflict of interest statement in the “Confidential to Editor” section, and submit your "Accept" recommendation.

Reviewer #1: All comments have been addressed

2. Is the manuscript technically sound, and do the data support the conclusions?

Reviewer #1: Yes

3. Has the statistical analysis been performed appropriately and rigorously? 

Reviewer #1: Yes

4. Have the authors made all data underlying the findings in their manuscript fully available?

Reviewer #1: No

5. Is the manuscript presented in an intelligible fashion and written in standard English?

Reviewer #1: Yes

6. Review Comments to the Author

Reviewer #1: (No Response)

7. PLOS authors have the option to publish the peer review history of their article (what does this mean?). If published, this will include your full peer review and any attached files.

Reviewer #1: No

---

## [Editor Report · Acceptance letter]

21 Apr 2022

PONE-D-20-34991R1 

A database study of clinical and economic burden of Invasive Meningococcal Disease in France 

Dear Dr. Marié:

I'm pleased to inform you that your manuscript has been deemed suitable for publication in PLOS ONE. Congratulations! Your manuscript is now with our production department. 

Kind regards, 

on behalf of

Dr Carla Pegoraro 

Staff Editor

PLOS ONE